# Antitumoral Effects of Tricyclic Antidepressants: Beyond Neuropathic Pain Treatment

**DOI:** 10.3390/cancers14133248

**Published:** 2022-07-01

**Authors:** Antonio Asensi-Cantó, María Dolores López-Abellán, Verónica Castillo-Guardiola, Ana María Hurtado, Mónica Martínez-Penella, Ginés Luengo-Gil, Pablo Conesa-Zamora

**Affiliations:** 1Facultad de Ciencias de la Salud, Universidad Católica de Murcia (UCAM), 30107 Guadalupe, Spain; aac17g@gmail.com (A.A.-C.); mariadolores.lopezabellan@outlook.es (M.D.L.-A.); penella73@hotmail.com (M.M.-P.); 2Servicio de Farmacia Hospitalaria, Hospital Universitario Santa Lucía, 30202 Cartagena, Spain; 3Grupo de Investigación en Patología Molecular y Farmacogenética, Servicios de Anatomía Patológica y Análisis Clínicos, Instituto Murciano de Investigación Biosanitaria (IMIB), Hospital Universitario Santa Lucía, 30202 Cartagena, Spain; veronicacgu.88@gmail.com (V.C.-G.); anah473@gmail.com (A.M.H.); 4Grupo de Investigación en Inmunobiología para la Acuicultura, Departamento de Biología Celular e Histología, Facultad de Biología, Universidad de Murcia, 30100 Murcia, Spain

**Keywords:** tricyclic antidepressants, antitumor therapy, imipramine, central nervous system, drug repurposing

## Abstract

**Simple Summary:**

Tricyclic antidepressants (TCAs) are old and known therapeutic agents whose good safety profile makes them good candidates for drug repurposing. As the relevance of nerves in cancer development and progression is being unveiled, attention now turns to the use of nerve-targeting drugs, such as TCAs, as an interesting approach to combat cancer. In this review, we discuss current evidence about the safety of TCAs, their application to treat neuropathic pain in cancer patients, and in vitro and in vivo demonstrations of the antitumoral effects of TCAs. Finally, the results of ongoing clinical trials and future directions are discussed.

**Abstract:**

Growing evidence shows that nerves play an active role in cancer development and progression by altering crucial molecular pathways and cell functions. Conversely, the use of neurotropic drugs, such as tricyclic antidepressants (TCAs), may modulate these molecular signals with a therapeutic purpose based on a direct antitumoral effect and beyond the TCA use to treat neuropathic pain in oncology patients. In this review, we discuss the TCAs’ safety and their central effects against neuropathic pain in cancer, and the antitumoral effects of TCAs in in vitro and preclinical studies, as well as in the clinical setting. The current evidence points out that TCAs are safe and beneficial to treat neuropathic pain associated with cancer and chemotherapy, and they block different molecular pathways used by cancer cells from different locations for tumor growth and promotion. Likewise, ongoing clinical trials evaluating the antineoplastic effects of TCAs are discussed. TCAs are very biologically active compounds, and their repurposing as antitumoral drugs is a promising and straightforward approach to treat specific cancer subtypes and to further define their molecular targets, as well as an interesting starting point to design analogues with increased antitumoral activity.

## 1. Introduction

High-throughput technologies and big data are allowing an unprecedented increase in the knowledge of molecular targets for the subsequent specific treatment of individual tumors. However, the time lapse between the identification of drug hits targeting these molecules to their final approval in the clinical practice is still a long process. For this reason, drug repurposing is becoming an efficient way to shorten this process, as already safe drugs are used for novel indications. In addition, it constitutes an interesting starting point to find analogues with improved properties for these new indications and, desirably, fewer effects on the initial ones. The use of tricyclic antidepressants (TCAs) in cancer treatment is not an exception to this paradigm, as they have proven, in vitro and in vivo studies, to display antitumoral activity, although the molecular rationale for these effects is not always fully understood. Although most studies have focused on central nerve system (CNS) tumors, the fact that all organs subjected to tumor origins are innervated supports the encouraging results in other cancers beyond the CNS. There was a traditional assumption that nerves played a passive role in tumorigenesis, as a mere vehicle for tumor invasion. However, in the past eight years, seminal studies have revealed an active role of nerves in cancer development and promotion [1]. This review focuses on the current evidence about the antitumoral effects of TCA, placing special attention on their central effects, the rationale for their direct antitumoral effects, and the promising results in clinical settings. In order to provide a whole picture of the pros and cons of TCA usage in this context, their safety and their central effects in oncology patients are also discussed.

### 1.1. The Role of Nerves in Cancer

Historically the involvement of nerves in cancer has been regarded as a mere vehicle for tumor cells to spread. Studies by Magnon et al. went a little further by demonstrating that neuron growth was increased in high-grade prostate cancers in comparison to low-grade or benign hyperplasia and concluding that catecholamines released by sympathetic nerves induce tumor growth, whereas cholinergic signaling, triggered by parasympathetic nerves, stimulated tumor spread [1]. Conversely, several studies have demonstrated, in different animal cancer models, that denervation produces a decrease in the development and promotion of cancer [2]. Given that nerves are part of the tissue microenvironment and release a myriad of neurotransmitters (NTs), hormones, and growth factors, it is not surprising that they play an active role in tumor biology, and, therefore, their modulation with neurotropic drugs can exert a beneficial effect against tumor progression. Increasing evidence supports the idea that tumor recapitulates part of the embryonic development or tissue regeneration by releasing neurotrophic growth factors to attract nerve terminals in a process called axonogenesis. In turn, nerves from the tumor microenvironment (TME) release NTs that activate stromal, immune, and endothelial cells from the TME modulating tumor proliferation, invasion, metastasis, and angiogenesis. The type of innervation (adrenergic, cholinergic, or sensory) or the NT released in a given tissue is critical for the outcome of nerve stimulation on a growing tumor [3]. For instance, activation of cholinergic signaling seems to inhibit progression of pancreatic cancer, whereas, in gastric cancer, parasympathetic nerves stimulate tumorigenesis and cancer stemness [4,5].

With all of this in mind, it is not surprising that growing evidence points to tricyclic antidepressants (TCAs) as important modulators of tumor development and potential antineoplastic drugs which can be repurposed.

### 1.2. TCA Mechanisms of Action

TCAs are amongst the first antidepressants being developed. They were discovered in the early 1950s. Their name refers to the chemical structure, which consists of three rings of atoms. TCAs act on approximately five different neurotransmitter pathways to achieve their antidepressant effects. The main mechanism of action of TCAs is binding and inhibition of the transporters responsible for the reuptake of the norepinephrine and serotonin (5-HT), resulting in the accumulation of these neurotransmitters in the presynaptic cleft. The increased concentrations of norepinephrine and serotonin in the synapse likely contribute to its antidepressant effect. However, TCAs have several modes of action other than the monoamine reuptake inhibition in the presynaptic cleft. They act also as competitive antagonists on postsynaptic alpha cholinergic (alpha1 and alpha2), muscarinic, and histaminergic receptors (H1), causing a variety of adverse effects, including dry mouth, confusion, hypotension, orthostasis, blurred vision, urinary retention, and sedation. Butryptiline, an analog of amitriptyline, has additional anti-H1 activity. Those TCAs predominantly inhibiting the reuptake of serotonin include clomipramine, imipramine, and trimipramine, whereas those predominantly inhibiting reuptakes of norepinephrine include desipramine (an imipramine metabolite), maprotiline, nortriptyline, and protriptyline. Finally, balanced reuptake inhibitors of serotonin and norepinephrine or unspecified inhibitors include amitriptyline, amitriptylinoxide, and amoxapine [6,7]. The formulas of the most used TCAs are shown in Figure 1.

An extensive search (>750 references) was been performed in the PubMed database by using the search terms “tricyclic antidepressant cancer”, with no language restrictions (data accessed in February 2022). The following inclusion criteria were used: (1) safety studies of TCA in oncology patients, (2) reports assessing the central effect of TCA in oncology patients, (3) in vitro and preclinical evidence of antitumoral properties of TCA, and (4) clinical evidence of antitumoral properties of antidepressant.

## 2. Safety and Central Effects of Antidepressant in Oncology Patients

### 2.1. Cancer Risk and Safety of TCAs

Prior to the use of repurposed drugs as antitumoral agents, it is crucial that they are deemed to be safe and well-tolerated. Given the dual role of nerves in cancer and the modulation of nerves exerted by antidepressants, several studies have evaluated whether TCAs increase cancer risk. This precaution not only applies to cancer incidence but also to prognosis. Research on animal models and tumor cell lines has highlighted several biological mechanisms that possibly support this association. However, epidemiological studies investigating cancer risk in patients receiving selective antidepressant such as serotonin reuptake inhibitors (SSRIs) and TCAs have yielded conflicting and inconclusive results.

Antidepressants, in general, and TCAs, in particular, did not seem to be associated with a higher incidence of colorectal cancer (CRC) [8,9]. In this line, the study conducted by Coogan et al. reported a significant reduction in the risk of colorectal cancer through the regular use of selective serotonin reuptake inhibitors, and the odds ratio did not differ by duration of use; however, they found a non-significant risk reduction for regular use of TCAs [10].

Regarding oral cancers, the prospective and nested case-control studies performed in the Taiwanese population by Chung et al. demonstrated the association between antidepressant use and decreasing the risk of oral cancer [11].

With regard to epithelial ovarian cancer (EOC), a case-control study in the Danish population that was conducted by Mørch et al. concluded that the use of SSRIs was associated with a decreased risk of epithelial ovarian cancer [12]. Moreover, eight case-control studies including 7878 EOC patients taking antidepressant versus 73,913 EOC patients with no antidepressant usage rendered a non-significant association with cancer risk [13].

Other similar studies in breast **[14,15,16,17]**, prostate [18,19], and gastric [20] cancer did not show any relationship between antidepressants’ use and increased cancer risk. These findings are not exclusive of tumor of epithelial origin, as current evidence on Non-Hodking Lymphoma (NHL) does not support the fact that antidepressants may promote or increase the risk of NHL in general or in specific common subtypes of NHL [21,22].

Regarding SNC tumors, the case-control study carried out by Walker et al. in 2011 found that tricyclic use may be associated with a subsequent reduction in the risk of glioma [23]. Subsequently, a nationwide case-control study of the association between long-term use of TCA and risk of glioma conducted by Pottegard et al. indicated that long-term use of TCAs was inversely associated with the risk of glioma, but this association was not statistically significant [24].

All of these findings indicate that TCAs do not increase the risk of cancer from different locations, with a few studies even demonstrating a reduction in tumor occurrence. 

### 2.2. Central Effects of TCA against Neuropathic Pain

Neuropathic pain is a common problem among oncology patients undergoing chemotherapy and other cancer treatments, such as surgery or radiotherapy, and it can also be attributed to the tumor presence causing nerve compression or invasion. Several drugs are used for pain relief in these patients: analgesics, anticonvulsive drugs, and antidepressants. In particular, the usefulness of tricyclic antidepressants (TCAs) in the treatment of neuropathic pain is a matter of growing interest.

Although antidepressants were not originally designed to act as analgesics [25], TCAs have been used as first-line therapy for the pharmacologic management of neuropathic pain for many years and in different diseases, including cancer [26,27]. However, the precise mechanisms underlying these analgesic effects remain unclear. In terms of clinical practice, the analgesic effects of these antidepressants for neuropathic pain manifest within a few days, while their antidepressant effects take 2–4 weeks to be observed, suggesting different modes of action [28].

Given the emergence of new drug treatments, clinical trials, and standards of quality for assessment of evidence, Finnerup et al. conducted a systematic review and meta-analysis of randomized double-blind studies of oral and topical pharmacotherapy for neuropathic pain. Their results led the authors to recommend the use of TCA as a first-line treatment in neuropathic pain, among other drugs, thus supporting the International Association for the Study of Pain Special Interest Group on Neuropathic Pain (NeuPSIG) guidelines [29].

Particularly in the cancer setting, there is a substantial amount of evidence about the utility of TCAs in patients with cancer-related neuropathic pain. In the 1980s, two main hypotheses about the role of TCAs in cancer pain were formulated. The first one suggests the effect of the TCAs in the emotional component of pain, and the other hypothesis suggest that TCAs themselves have a specific analgesic action linked to a direct activity on the structures of the central nervous system [30]. In this line, there are several chronic pain diseases, such painful diabetic neuropathy, migraine headache, and mixed tension–vascular, in which TCA drugs have shown an analgesic effect independent of antidepressant effects by using lower doses than for depression treatment [31].

More recent evidence for the possible mechanism of action of the analgesic effect comes from several studies on animal models [32]. These studies demonstrated that norepinephrine plays a crucial role in the inhibition of neuropathic pain. In fact, the increased concentrations of norepinephrine in the spinal cord caused by TCAs inhibit neuropathic pain through α_2_-adrenergic receptors. Dopamine and 5-HT also increase in the central nervous system and may enhance the analgesic effects of norepinephrine to inhibit neuropathic pain [32].

In a prospective randomized double-blind placebo-controlled study with 120 cancer patients experiencing severe neuropathic cancer pain, amitriptyline demonstrated to be effective in relieving neuropathic pain and other neuropathic symptoms; however, pregabalin and gabapentin provided a significantly higher performance [33]. Amitriptyline application for cancer pain treatment was also tested in a previous prospective study enrolling 818 patients, confirming that neuropathic cancer pain can be relieved by multimodal treatment including amitriptyline, following the World Health Organization (WHO) analgesic ladder, a strategy proposed by the WHO to provide adequate pain relief for oncology patients [34]. Amitriptyline also showed similar efficacy to SSRIs when treating cancer pain and other painful syndromes with deafferentation component [35]. In another study, 44 patients with chemotherapy-induced neuropathic symptoms were treated with low-dose amitriptyline or a placebo during eight weeks, and amitriptyline did not seem to improve sensory neuropathic symptoms; however, there was a trend to improve the quality of life in the amitriptyline group [36].

In 2010, Arai et al. observed that a combination of low-dose gabapentin and imipramine significantly decreased neuropathic pain and paroxysmal pain episodes in cancer patients, without severe adverse effects [37].

There is also evidence of the effectiveness of mirtazapine, a drug usually used to treat depression and sometimes obsessive–compulsive and anxiety disorders, to improve multiple central symptoms, including pain, nausea, anxiety, insomnia, and appetite in advanced cancer patients [38,39].

A recent review of the up-to-date guidelines of ESMO (European Society for Medical Oncology), ASCO (American Society of Clinical Oncology), ONS (Oncology Nursing Society), NCI (National Cancer Institute), and NCCN (National Comprehensive Cancer Network) concluded that TCAs are recommended as first-line treatment of chemotherapy-induced peripheral neuropathy, along with other antidepressants and opioids [40]. However, there is no consensus about the ideal therapeutic agent, so the treatment of cancer-related neuropathic pain remains a matter of interest and research.

The benefit of TCAs in oncology patients is not new, as decades ago, preliminary findings suggested that cancer patients with major depression could benefit from antidepressant treatment [41]. In 1998, a controlled trial of fluoxetine and desipramine in depressed women with advanced cancer suggested that both drugs were effective and well-tolerated in improving depressive symptoms and quality of life in women with advanced cancer, acknowledging the need for confirmation in a larger sample [42]. Several small studies have shown the efficacy of amitriptyline in this regard, though TCAs tend to produce a higher rate of side effects compared to other antidepressants such as paroxetine [43,44].

However, in a systematic review and meta-analysis reevaluating the role of antidepressants in cancer-related depression, the use of amitriptyline or desipramine did not lead an improvement in depression, and the authors noted the need to perform randomized trials to identify optimal treatments for managing cancer-related depression [45]. Raddin et al. observed that depression and life quality improved with mirtazapine but again concluded that evidence-based pharmacologic treatments for depression in cancer patients are needed [46]. Furthermore, a Cochrane Database Systematic Review assessing the efficacy, tolerability, and acceptability of antidepressants for treating depressive symptoms in cancer patients concluded that there is very low certainty evidence for the effects of antidepressants, including TCAs, compared with a placebo [47]. A possible explanation for this poorly consistent evidence is that studies evaluating the antidepressants’ usefulness in cancer patients with major depression are often affected by intercurrent disease and several treatment variables, making it challenging to conduct large-scale studies. Here, we highlight the need for large randomized trials on this matter.

Clinical trials in which TCAs were used to treat neuropathic pain in oncology patients are shown in Table 1.

## 3. In Vitro and Preclinical Evidence of Antitumoral Properties of Tricyclic Antidepressants

Experimental studies using cell culture and animal models are crucial to understand the possible mechanism of action and the molecular insights associated with the effects of TCAs as antitumoral drugs. In addition to their classical effects on neurotransmission, recent studies in depression disorders suggest that TCAs can also suppress microglial activation by the inhibition of the production of proinflammatory cytokines such as TNF-α, IL-1β, and IL-6 by glial cells. Antidepressants were also found to promote resilience of injured neurons during neuroinflammatory conditions in vivo. [52,53,54] Taking into account these studies, and since inflammation is a critical component of tumor progression, it is possible that treatment with TCAs might prevent or reduce the risk of developing cancer in the long term. Table 2 shows the most relevant molecular properties of TCAs in cancer. In addition, as most of the antitumoral studies have been carried out by testing imipramine, the most relevant molecular pathways involved in its antitumoral effects are shown in Figure 2. Since these effects might be tumor dependent, in this section, we review preclinical studies carried out according to the cancer type or location.

**Table 2 cancers-14-03248-t002:** Relevant in vitro and preclinical evidence of antitumoral properties of tricyclic antidepressants.

Tumor	Compound	Targets of TCA	Effects	Reference
Glioblastoma	Imipramine	Extrinsic/intrinsic pathways and suppression of ERK/NF-κB signaling.	Induction of apoptosis.	[55]
		Inhibition of yes-associated protein (YAP), independent of Hippo pathway.	Suppression of tumor proliferation. Reduced orthotopic tumor progression and prolonged survival of tumor-bearing mice.	[56]
		PI3K/Akt/mTOR signaling.	Autophagic cell death.	[57]
	Amitriptyline Imipramine	p65 NF-κB expression.	Partially reversion of mitochondrial abnormalities.	[58]
		Silencing of the glioma stem cells’ profile.	Partially reversion of the malignant phenotype.	[59]
	Imipramine blue followed by liposomal doxorubicin	Profilin-1, scinderin, α-actin, calgranulin, and RhoGDP dissociation inhibitor α.	Reduction in actin fiber Formation.	[60]
	Imipramine + ticlopidine	Imipramine activates adenylate cyclase and induces cAMP-mediated autophagy.	By elevating cAMP levels via distinct mechanisms, combined therapy increased autophagic flux.	[61]
	Clomipramine, norclomipramine, amitriptyline, and doxepin	Potent inhibitors of cellular respiration. Inhibition of complex III of the mitochondrial respiratory chain.	Increasing cell death.	[62]
	Clomipramine + imatinib	Inhibits complex-III of the respiratory chain, resulting in increased ROS, cytochrome C release and caspase-activated apoptosis.	Inhibition of cell growth and enhanced cell death. Synergistic apoptosis. There was also a synergistic effect in autophagy by the combination.	[63]
Sonic Hedgehog Medulloblastoma	Imipramine blue in liposomal nanoparticle (liposome–IB)	NADPH oxidase (NOX) family.	Dose-dependent decrease in SHH MB cell viability and migration. Inhibition of tumor growth. Reduced tumor volume. Complete tumor response. Improved survival.	[64]
Neuroblastoma	Imipramine	Potentiates ER-stress-induced death of SH-SY5Y cells.	Concentration-dependent reduction of the relative viability.	[65]
	Clomipramine + vinorelbine	Capable of potentiating vinorelbine cytotoxicity. Leads to ROS production through inhibition of complex III of the respiratory chain, resulting in increased ROS, mitochondria damage, cytochrome C release, and caspase- activated apoptosis of tumorigenic cell lines.	Increased the percentage of apoptotic cells.	[66]
Breast cancer	Imipramine blue + nanoparticle-based delivery approach	Inhibition of FoxM1.	Blockage of the ability of repair DNA strand breaks by homologous recombination (HR).	[67]
	Amitriptyline	Unknown.	Reduced viability.	[68]
	Clomipramine	Inhibition of distinct ubiquitin E3 ligases. Specifically blocks ITCH auto-ubiquitylation, as well as p73 ubiquitylation.	Reduces cancer cell growth and synergizes with gemcitabine or mitomycin in killing cancer cells by blocking autophagy.	[69]
	Imipramine	Able to cause changes in the structural organization of the phosphatidylserine bilayer and that these changes correlate with their MDR-reversing activity and potency to inhibit PKC.	Inhibition of either the cell growth or protein kinase C (PKC) in MCF7 and P338 doxorubicin resistant cells.	[70]
Colorectal cancer	Imipramine	Fascin1 inhibition.	Dose-dependent anti-invasive and antimetastatic activities.	[71]
Head and neck squamous cell carcinoma	Imipramine blue	Inhibition of Twist1-mediated let-7i downregulation and Rac1 activation and the EMT signaling.	Represses mesenchymal-mode migration in two-and-a-half-dimensional/3D culture system	[72]
Lung Cancer	Imipramine	EGFR/PKC-δ/NF-κB pathway suppression in non-small-cell lung cancer.	Induced apoptosis of NSCLC cells via both intrinsic and extrinsic apoptosis signaling. DNA damage increased. Invasion and migration of NSCLC cells suppressed by imipramine.	[73]
	Amitriptyline	Increases death receptor (DR) 4 and 5 expression, a requirement for TRAIL-induced cell death.	Blockage of autophagy by inhibiting the fusion of autophagosomes with lysosomes.	[74]
	Desmethylclomipramine	Inhibits in vitro the E3 ubiquitin ligase Itch.	Inhibits lung cancer stem cells’ growth, decreases their stemness potential, and increases the cytotoxic effect of conventional chemotherapeutic drugs.	[75]
Acute myeloid leukemia	Imipramine blue + pimozide	Induces calcium release from the ER/lysosomes and can inhibit tyrosine phosphorylation of STAT5.	Important calcium channel blocker activity converging with IB on mitochondrial oxidative metabolism.	[76]
Lymphoma	Imipramine blue	Inhibition of NADPH oxidase NOX4 in Burkitt lymphoma.	Potent growth inhibition.	[77]
	Clomipramine	SERT-binding (SERT/SLC6A4)	Promoted growth arrest of chronic lymphocytic leukemia (CLL), Small lymphocytic lymphoma (SLL), mantle cell lymphoma (MCL), follicular lymphoma (FL), and diffuse large B cell lymphoma (DLBCL).	[78]
	Imipramine dimers	Inhibition of the human serotonin transporter (hSERT).	Induction of cell death.	[79]
Bladder cancer	Clomipramine	Inhibition of distinct ubiquitin E3 ligases. Specifically blocks ITCH auto-ubiquitylation, as well as p73 ubiquitylation.	Reduces cancer cell growth and synergizes with gemcitabine or mitomycin in killing cancer cells by blocking autophagy.	[69]
Prostate cancer	Imipramine	Suppression of AKT and NF-κB-related signaling proteins and secretion of tumor necrosis factor-α (TNF-α), interleukin-1β (IL-1β), and monocyte chemoattractant protein-1 (MCP-1).	Attenuated cell viability, migration, and invasion.	[80]
		Eag1 channel protein expression.	Inhibition of the flow thought the channel.	[81]
	Clomipramine	Inhibition of distinct ubiquitin E3 ligases. Specifically blocks ITCH auto-ubiquitylation, as well as p73 ubiquitylation.	Reduces cancer cell growth and synergizes with gemcitabine or mitomycin in killing cancer cells by blocking autophagy.	[69]
		Inhibition of autophagy.	Effective in inhibiting autophagy and enhanced therapeutic response in ENZA-resistant cells in vitro and in vivo, using the orthotopic xenograft model combined with ENZA.	[82]
Melanoma	Amitriptyline, nortriptyline, and clomipramine	Inhibition of complex III of the mitochondria has been postulated as a mechanism of action.	All three agents showed increasing inhibition with increasing concentration in both cell lines and primary cell cultures.	[83]
	Imipramine	Ether à go-go (hEAG) channels and Ca^2+^ -activated channels (K_Ca_) of the IK/SK type.	Increasing concentrations of imipramine reduced the proliferation of IGR1 cells.	[84]
Hepatocellular carcinoma	Amitriptyline	Inhibition of β-catenin and Ki-67.	Decreases β-catenin-induced liver enlargement in zebrafish. Decreases tumor burden in a mouse HCC model. Amitriptyline treatment significantly decreases tumor cell proliferation, due to a reduction in the amount of Ki-67.	[85]
	Desipramine	Inhibition of the phosphorylation of ERK1/2, JNK, and p38.	Increases ROS generation and cell death in a dose-dependent manner. Loss of mitochondrial membrane potential.	[86]
Osteosarcoma	Desipramine and Nortriptyline	Calcium homeostasis;	Causes a rapid and sustained rise of intracellular Ca^2+^ in a concentration-dependent manner.	[87,88]
	Desipramine	p38 MAPK-associated activation of caspase 3.	Causes Ca^2+^-independent apoptosis.	[89]
Multiple myeloma	Amitriptyline	Decreases histone deacetylases’ expression and inhibits their activity (HDAC3, -6, -7, and -8). Induces p53, activates caspase 3, and decreases antiapoptotic Bcl-2 and Mcl-1 in tumor tissues.	Amitriptyline induces cell apoptosis. Oral administration decreases tumor growth in two MM xenograft models derived from murine and human cells.	[90,91]
	Nortriptyline	Most likely the target would be organic cation transport machinery.	Dose- and time-dependent toxicity on cells. Arrests cell cycle at G2/M phase. Causes mitochondrial membrane depolarization. Increases caspase-3 activity. Induction of apoptosis.	[92]

### 3.1. Cancer of Central Nervous System

The high concentration reached in the CNS by TCAs offers an interesting approach to exploit these drugs for treating tumors from this location, which is not always accessible for therapeutic agents. In fact, several in vitro and preclinical studies and reviews have evaluated the effect of antidepressants in tumors of the central nervous system. The review by Abadi et al. suggested that the high occurrence rates of depression in glioblastoma multiforme patients, as well as the overlap of molecular and cellular mechanisms involved in the pathogenesis of these diseases, make antidepressants with antitumor effects an affordable strategy for the treatment of this tumor [93]. Some of the in vitro and preclinical studies go back to the end of the twentieth century, with the one reported by Richelson in 1978 being the earliest. Briefly, the author showed that antidepressants were potent competitive inhibitors of histamine H1 receptors in mouse neuroblastoma cells [94]. Afterward, in 1981, Albouz et al. found that TCAs decreased sphingomyelinase activity in murine neuroblastoma and human fibroblast cell cultures, while other lysosomal enzymes were not modified [95]. Furthermore, the results of the study by Nakaki et al. suggested that imipramine binding sites were present together with the 5-HT uptake sites in NCB-20 cells, and those sites interacted functionally but were different biochemically [96]. The study performed by Ogata N. et al. showed that imipramine blocked sodium, calcium, and potassium channel currents in a reversible and concentration-dependent manner, which could play a role in the clinical effect [97]. In fact, imipramine potentiates mitochondrial dysfunction and ER-stress-induced death [65], possibly due to an alteration of ionic homeostasis. The study by Carignani et al. analyzed the effect of desipramine and imipramine in SK3 channels expressed in human medulloblastoma, showing a complete, reversible, and concentration-dependent block [98]. More recent studies support the role of TCAs as adjuvant therapy for cancer of the CNS. Bielecka-Wajdman et al. reported that antidepressant drugs, particularly imipramine and amitriptyline, stimulated the switching phenotype from glioma stem cells to non-glioma stem cells, which partially reverse the malignant phenotype of glioblastoma multiple [59].

Other studies, such as that by Slamon et al., suggested an effect of antidepressants in cell death. They analyzed the effects of acute exposure to selected antidepressants on DNA damage in cultured C6 rat glioma cells, showing that the antidepressants induced significant amounts of DNA damage in C6 cells [99]. Several years later, Bilir et al. demonstrated in the same cell line (C6) that the cytotoxic effect of chlorimipramine could be potentiated by combination with imatinib, suggesting the potential clinical application of the combination treatment [63]. Along this same line, the study by Qi et al. showed that desipramine induced apoptosis by upregulating caspase 3 gene expression and disturbing homeostasis in the calcium signaling system [100]. Moreover, the study by Levkovitz et al. strongly suggested that selected antidepressants induced apoptosis in neuronal and glial cell lines, with a high sensitivity to cancer cells compared with primary brain tissue [101]. Furthermore, Bilir et al. showed that both clomipramine and lithium chloride seemed to potentiate the cytotoxicity induced by vinorelbine [66]. Moreover, Ma et al. showed that desipramine induced typical apoptotic morphology of chromatin condensation in rat glioma C6 cells and activated intracellular caspase 9 and caspase 3, with no change in mitochondrial membrane potential [102]. 

A series of studies have reported that TCAs increase autophagic death of glioma cells. Ma et al. indicated that desipramine could induce autophagy through the PERK-ER stress pathway in C6 glioma cells, thus providing new insights into another potential benefit of desipramine in the adjuvant therapy of cancer [103]. Moreover, Shchors et al. demonstrated that imipramine increased autophagy and therapeutic benefits in tumor-bearing animals; this effect was further increased by the anticoagulant ticlopidine via reducing cell viability in culture [61]. Likewise, the study by Jeon et al. suggested that imipramine exerted antitumor effects on PTEN-null U-87MG human glioma cells by inhibiting PI3K/Akt/mTOR signaling and by inducing autophagic cell death but not apoptosis [57]. In contrast, Hsu et al. demonstrated that imipramine was a potential anti-glioblastoma drug which induced apoptosis and had the capacity to inhibit ERK/NF-κB signaling [55]. 

Some studies suggest that stressed mitochondria represent a new target for CNS cancer therapy. Higgins et al. performed an investigation about the effects of TCAs on cellular respiration, and the results showed that clomipramine was a potential antineoplastic agent for targeting the mitochondria of glioma cells [62]. The study by Bielecka-Wadjman et al. concluded that imipramine partially reversed glioblastoma multiforme abnormalities by restoring a proper function of mitochondria [58].

Other diverse mechanisms of tricyclic antidepressants in CNS tumors have been proposed. The findings by Zhu et al. indicated that desipramine exerted complex gradually evolving effects on the expression of tyrosine hydroxylase protein (decreases) and tyrosine hydroxylase mRNA (increases), possibly in response to increased synaptic availability of norepinephrine [104]. Moreover, the results by Hisaoka et al. showed that amitriptyline acutely increased CREB activity in PTK- and ERK-dependent manners, which might contribute to the expression of certain genes, including glial-cell-line-derived neurotrophic factor (GDNF) in glial cells [105]. A subsequent study also conducted by Hisaoka et al. showed that G protein signaling was involved in amitriptyline-evoked GDNF production in rat C6 astroglial cells in a study that sought to elucidate the mechanism of amitriptyline-induced production of GDNF in astroglial cells [106]. The study by Munson et al. showed that nano-imipramine blue enhanced the efficacy of nano-doxorubicin chemotherapy, demonstrating the promise of an anti-invasive compound as an adjuvant treatment for glioma [60]. Recently, the study by MacDonald et al. concluded that liposome–imipramine blue is a potential novel nanoparticle-based therapeutic for the treatment of Sonic Hedgehog medulloblastoma [64]. Along the same line, the review about the reuse of molecules for glioblastoma therapy by Koehler et al. included IB as a candidate molecule as it limits cancer migration and decreases inhibition of transcript factors known to aid cell survival [107]. Furthermore, Lucki et al. suggested that dexamethasone, in the presence of desipramine, enhanced MAPK/ERK1/2 signaling, a key player in neural plasticity and neurogenesis processes that is impaired in major depression disorder [108]. Along the same line, the data obtained by Lieberknecht et al. indicated that mirtazapine and imipramine had neuroprotective effects against H_2_O_2_-induced cell death present in depressed patients [109]. Recently, the study by Zhang et al. showed that fascin expression in glioma tissue was higher than that of normal brain tissue and that high fascin expression correlated with World Health Organization (WHO) grading of glioma patients. Moreover a multivariate analysis showed that high expression of fascin protein was an independent predictor of the prognosis of patients with glioma [110]. Intriguingly, Wang et al. showed that the TCA with anti-fascin activity, imipramine, significantly retarded the proliferation of primary and immortalized glioma cells by inhibiting YAP protein, a recognized oncogene in glioma [56].

### 3.2. Breast Cancer

A study in 2016 [67] suggested that IB suppresses breast cancer growth and metastasis, both in vitro and in preclinical mouse models. The effect was allegedly due to the inhibition of the ability of breast cancer cells to repair DNA, but without affecting normal mammary epithelial cells. This antitumoral effect is caused by the inhibition of Forkhead box M1 (FOXM1), a multifunctional transcriptional oncoprotein that is involved in cell proliferation and DNA repair. These findings highlight the potential of IB as a safe regimen for treating breast cancer patients. Given that FOXM1 is an established therapeutic target for several cancers, the identification of a compound that inhibits FOXM1- and FFOXM1-mediated DNA repair has important translational potential for treating many aggressive cancers. Very recently, Timilsina et al. demonstrated that imipramine triggers cell cycle arrest in triple-negative (TNBC) and estrogen-receptor-positive (ER+) breast cancers and by blocking heightened DNA double-strand breaks repair machinery. As a result, imipramine decreases the expression of cell-cycle and DNA repair proteins. More specifically, this TCA inhibits the growth of ER+ cancers by disrupting the estrogen receptor- α (ER-α) signaling and sensitizes TNBC to the PARP inhibitor Olaparib, using mouse models and ex vivo explants from breast cancer patients [111]. Despite the fact that fascin is overexpressed in TNBC [112,113,114], there are no studies evaluating the action of imipramine as an anti-fascin agent. In the case of amitriptyline, the viability of MCF-7 and SK-BR-3 cells exposed to 50–100 μM was markedly inhibited, but the exact mechanism is unknown [68]. Old studies dealing with membrane interactions performed by Pajeva et al. described a significant correlation between the MDR-reversing activity of imipramine and other drugs in doxorubicin-resistant MCF-7 tumor cells. These authors found that imipramine interacts with artificial membranes composed of phosphatidylcholine or phosphatidylserines [70].

### 3.3. Colorectal Cancer

Although advances have been made in screening, early detection, and management of colorectal cancer, therapeutic innovations have been scarce [115]. However, some widely used TCAs, such as imipramine, desipramine, and amitriptyline, have emerged as possible promising repurposing drugs in colorectal cancer [116]. Following this line, in 2020, Alburquerque-Gonzalez et al. showed that imipramine attenuated cell migration and invasion of colorectal cancer cells by inhibition of fascin [71], a key protein in actin bundling that plays a causative role in tumor invasion and is overexpressed in different cancer types with poor prognosis [117]. This was the first study that demonstrated an antitumoral role of imipramine as a fascin inhibitor [118]. Its anti-invasive and anti-cell-migration properties make imipramine an interesting drug candidate to halt metastasis and reduce tumor progression, not only against colorectal cancer but also against other invasive tumors; for example, cytokine-induced fascin expression is regulated by Signal Transducers and Activators of Transcription 3 (STAT3) and is required for breast cancer cell migration [119]. Pathways such PKCδ and Wnt-1, which lead to STAT3α activation, lead to the upregulation of fascin expression also in breast cancer [120]. Some transcription factors, such as nuclear factor κB (NFκB) and hypoxia-inducible factor1 (HIF1A), are also able to promote fascin gene transcription in gastric cancer and pancreatic ductal adenocarcinoma [121,122]. Moreover, the in vitro study carried out in colorectal cancer cells by Kabolizadeh et al., the authors concluded that desipramine augmented the cytotoxicity of the platinum drug, thus showing a synergistic effect [123].

### 3.4. Head and Neck Cancer

Since the major route for dissemination of head and neck squamous cell carcinoma (HNSCC) is local invasion rather than distant metastasis, targeting the locally invasive cancer cells is more important than preventing systemic metastasis. Under this premise, in 2016, Yang et al. demonstrated that treatment with imipramine blue (IB), a novel analogue of antidepressant imipramine normally used as a NADPH-oxidase (NOX) inhibitor, halts head and neck cancer invasion through the inhibition of the epithelial–mesenchymal transition (EMT), a process considered crucial in tumor progression during which cells lose apical–basal cell polarity and gain motility [72]. IB promotes degradation of the EMT inducer Twist1 and reduces radical oxygen species production, which inactivates the NF-κB pathway in vitro and in vivo. Taken together, these results suggest that IB is a potent EMT inhibitor and demonstrate the anti-invasive mechanisms of this drug in HNSCC.

### 3.5. Lung Cancer

Accumulating evidence from in vitro and early preclinical studies shows that imipramine might play an important antitumoral role in small cell lung cancer (SCLC). In fact, in 2013, Jahchan et al., after carrying out a methodical bioinformatic analysis, identified imipramine as a potential candidate to consider for drug repurposing in SCLC [124]. According to these authors, treatment with imipramine not only produced cytotoxic effects in SCLC cell lines but also controlled SCLC tumor growth in animal models (NSG mice). An in vivo analysis of the endogenous SCLC tumors developing in the lungs of Rb/p53/p130–mutant mice after 30 days of treatment with imipramine revealed that imipramine-treated mice had fewer and smaller SCLC tumors than control mice. In that case, imipramine induced apoptosis in both chemonaive and chemoresistant SCLC cells in culture and in human SCLC tumors transplanted into immunocompromised mice. Cell death was induced, in part, by disrupting autocrine survival signals involving neurotransmitters and their G-protein-coupled receptors. In the same way, tricyclic drugs inhibited the growth of other neuroendocrine tumors, such as pancreatic neuroendocrine tumors and Merkel cell carcinoma. As a result, the authors concluded that imipramine and similar TCAs could potentially be used as a second-line therapy in cisplatin/etoposide refractory SCLC. Based on these preclinical models, Lothian et al. carried out a retrospective study including 876 stage-four SCLC patients to analyze the association between the use of TCA and an improvement in the overall survival. Although the authors concluded that the results of the preclinical models were not reproduced in their clinical cohort and suggested that preclinical data should be treated with caution before application in the clinic, in their study, only 5 patients out of 876 received TCA [125]. On the other hand, for non-small-cell lung cancer (NSCLC), the in vivo and in vitro studies performed by Yueh et al. reported that imipramine was able to induce intrinsic and extrinsic apoptosis and reduced the invasion and the migration potential of NSCLC cells. In this case, the triggers of this antitumor effect seemed to be the increase in DNA damage and the decrease in phosphorylation of EGFR/PKC-δ/NF-κB and their downstream proteins [73]. In this line, the study performed by Zinnah et al. showed that amitriptyline could sensitize TRAIL-resistant A549 lung cancer cells to tumor-necrosis-factor-related apoptosis-inducing ligand (TRAIL) and enhance TRAIL-induced apoptosis through DR4 and DR5 upregulation and autophagy inhibition [74]. Finally, the treatment of lung cancer stem cells with desmethylclomipramine (DMCI), the active metabolite of clomipramine, showed growth reduction and enhanced the cytotoxic effect of chemotherapeutic drugs. This effect is due to the inhibition of the E3 ubiquitin ligase Itch by DMCI, with the expression of E3 ubiquitin ligase being a negative prognostic factor in different lung tumors [75].

### 3.6. Acute Myeloid Leukemia

IB has been newly investigated as a disruptor of calcium regulation in acute myeloid leukemia (AML) cells. At concentrations of 75–150 nM, IB induced selective apoptosis for FLT3-ITD+ cells by increasing cytosolic and lysosomal Ca^2+^ release [76,126]. Thus, treatment with IB against FLT3-ITD+ cells might provide new hope for refractory AML patients.

### 3.7. Lymphoma

Since the serotonin transporter (SERT/SLC6A4) is expressed in a wide range of B cell lines of diverse neoplastic origin, cell growth arrest in response to SERT ligands, including the antidepressants clomipramine and fluoxetine, is possible. Following this assumption, Chamba and colleagues demonstrated, on ex vivo patient cells, that clomipramine is capable of promoting growth arrest in chronic lymphocytic leukemia (CLL), small lymphocytic lymphoma (SLL), mantle-cell lymphoma (MCL), follicular lymphoma (FL), and diffuse large-B-cell lymphoma DLBCL [78]. Ethylene glycol N-interlinked imipramine dimers of various lengths synthesized by Bright et al. showed a potent inhibition of cellular viability, inducing cell-type-specific death mechanisms in a chemoresistant Burkitt’s lymphoma cell line. In contrast to clomipramine and fluoxetine, imipramine dimers seems to be moderate inhibitors of the human serotonin transporter SERT, while the induction of cell death occurred independently of SERT expression [79]. Finally, in xenografts of the chick chorioallantoic membrane (CAM), Klingenberg et al. demonstrated a potent growth inhibition of Burkitt’s lymphoma, using IB at 10 µM. It is not yet clear whether the observed effects are primarily based on NOX4 inhibition or whether off-target effects are involved [77].

### 3.8. Bladder Cancer

Experiments performed by Rossi M. et al. [69] showed that clomipramine reduces cell growth and induces cell death of HT-1376 bladder cancer cells. Moreover, the combination of clomipramine and gemcitabine seems to be synergistic. An unexpected result found by these authors was the identification of clomipramine as an E3 inhibitor which blocked ITCH auto-ubiquitylation, as well as an inhibitor of ITCH-dependent ubiquitylation of p73 [69].

### 3.9. Prostate Cancer

Emerging research shows that TCAs such as imipramine [80] and desipramine [127] attenuate cell viability, migration, and invasion of prostate cancer cells (PC-3) in vitro. Söğüt et al. found that imipramine was able to reduce currents, conductivity, and protein expression on the Eag1 channel in prostate cell line DU145. [81] The use of clomipramine in prostate cancer cell lines resulted in increased apoptosis. In vivo experiments with mice implanted with ENZA-resistant cells showed that the combination of ENZA and clomipramine was able to reduce tumor growth [82].

Despite these promising results, the available experimental evidence for TCAs in prostate cancer is still scarce compared to other standard antitumor agents.

### 3.10. Melanoma

There are few studies about the effects of TCA in melanoma. In 2002, it was shown that increasing concentrations of imipramine reduced the proliferation and accumulation of IGR1 melanoma cells in the G0/1 phase. Moreover, human ether à go-go (hEAG) potassium channels, which enhance cell proliferation, were sensitive to inhibition by imipramine in in vitro melanoma cells, thus decreasing proliferation [84]. In primary cell culture from metastatic melanoma, nortriptyline showed higher activity than clomipramine or amitriptyline, but in either case, the three TCA tested in vitro in primary cell culture and in two melanoma cell lines showed activity against melanoma [83].

### 3.11. Hepatocellular Carcinoma

Hepatocellular carcinoma (HCC) is one of the most lethal human cancers. The search for effective treatments is urgently needed. Evason et al., with the aim of searching targeted treatments in HCC, developed a new transgenic model in zebrafish that mimics the HCC subtype characterized by the presence of activating mutations in the *CTNNB1* gene encoding β-catenin. A total of 8 compounds out of 960 drugs tested were able to suppress β-catenin-induced liver enlargement; two of them were antidepressants, a TCA and an SSRI. Moreover, the authors demonstrated that amitriptyline decreased tumor burden in a mouse HCC model [85]. Finally, an in vitro study showed that desipramine increases ROS generation and cell death in a dose-dependent manner. This mechanism appears to be triggered by a loss of mitochondrial membrane potential mediated by an increase in intracellular calcium levels, leading to activation of MAPK signaling and promoting antiproliferative effects in Hep3B HCC cells, ending with an apoptotic cell death [86].

### 3.12. Osteosarcoma

In human osteosarcoma MG63 cells, it was shown that desipramine and nortriptyline caused a rapid and sustained increase of intracellular calcium in a concentration-dependent manner in osteoblasts. The unregulated elevation of intracellular calcium caused by the high concentrations of these TCAs showed a cytotoxic effect in MG63 cells [87,88]. In this cell line, there are results which suggest that desipramine causes calcium-independent apoptosis by inducing p38 MAPK-associated activation of caspase 3 [89].

### 3.13. Multiple Myeloma

Drug repurposing has been proved to be an effective strategy to meet the urgent need for novel anticancer agents for multiple myeloma (MM) treatment. The TCA nortriptyline has shown greater inhibitory and apoptotic effects in U266 MM cells than cisplatin; however, the cisplatin–nortriptyline combination also indicated strong antagonism [92]. Ten years ago, in vitro and in silico studies demonstrated that amitriptyline showed anti-myeloma activity by inducing cell apoptosis through the inhibition of histone deacetylases [90]. In vivo studies confirmed that amitriptyline has anti-myeloma activity, thus decreasing the tumor growth and increasing the overall survival in two MM xenograft models derived from murine and human MM cells. An in vitro study combined amitriptyline with bortezomib and synergistically induced MM cell apoptosis [91].

## 4. Clinical Evidence of Antitumoral Properties of Antidepressants

Despite the growing evidence presented about the in vitro and preclinical studies demonstrating the antitumoral properties of TCAs, there are very few clinical studies aiming to confirm these effects. Most of them are simply descriptive and retrospective studies where an association between TCAs and prognosis, recurrence risk, or survival is evaluated. 

In this context, the results obtained by Zingone et al. in patients from the NCI-Maryland lung cancer study showed that the use of TCAs or norepinephrine–dopamine uptake inhibitors (NRDIs) was associated with extended lung-cancer-specific survival, which was maintained after adjustment for the clinical indication for these drugs, thus suggesting direct effects on lung cancer biology. Those results suggested that the evaluation of antidepressants as adjunct therapeutics with chemotherapy may have a translational effect for lung cancer patients [128]. However, the study by Abdel Karim et al. did not reveal any significantly positive impact of antidepressants on the overall survival of the studied cohort of lung cancer patients [129]. Along these lines, a study which aimed to determine whether the anticancer action of TCAs translated to improved survival in patients with colorectal cancer or glioma showed no significant mortality reduction [130].

In regard to prognosis, the meta-analysis by Chen et al. found that the use of antidepressants in CRC after diagnosis is common and was not associated with mortality [131]. According to the survey by Pocobelli et al., the use of antidepressants was not associated with a risk of recurrence [132]. These findings were similar to those found in breast cancer, wherein SSRIs and TCAs were not associated with the risk of recurrence or mortality [133].

Regarding experimental studies in cancer patients, Table 3 shows the clinical trials evaluating antitumoral effects of TCAs. At present, most of them are ongoing, and only one trial has reported results concluding that desipramine does not imply clinical or radiographic improvement.

## 5. Limitations of the Study

The mechanisms by which TCAs induce tumor cell death seem to be heterogeneous and dependent on tumor type and even subtype. Given this fact and that of the limited number of clinical trials evaluating the antitumoral effects of TCAs, it is not surprising that the promising results obtained in in vitro and in vivo preclinical studies are not supported by the clinical results. In fact, one phase 2a clinical trial using desipramine in small cell lung cancer and other high-grade neuroendocrine tumors was terminated after finding no clinical or radiographic benefit. Despite this fact, new clinical trials are being initiated in different cancer types, evaluating the effect of various TCAs. More basic and translational research will be needed to gain a more detailed understanding of the mechanisms of action and to validate molecular targets of TCAs. As this knowledge continues growing, the selection of patients according to specific histological or molecular features seems crucial for finding the benefit of TCAs as antitumoral agents.

## 6. Conclusions

Different lines of evidence show the therapeutic action of TCAs in oncology patients, and not only restricted to cancers of the CNS. These treatments do not seem to increase cancer risk, and the benefit is far beyond the effect on neuropathic pain as; according to in vitro and preclinical studies, it also includes direct antitumoral effects. However, there are still too few experimental studies in the clinical setting to confirm cell culture and animal model assays, thus reinforcing the necessity for translating this knowledge to the patient context. TCAs will probably not be beneficial to all types of cancer. The different involvement of nerves depending on the tumor subtype warrants future studies to discern which clinical, histological, or molecular features are associated with good antitumor response of the TCAs. At the same time, important research fields will be the discovery and characterization of TCA analogues with less central effects and increased antitumoral activity.

## Figures and Tables

**Figure 1 cancers-14-03248-f001:**
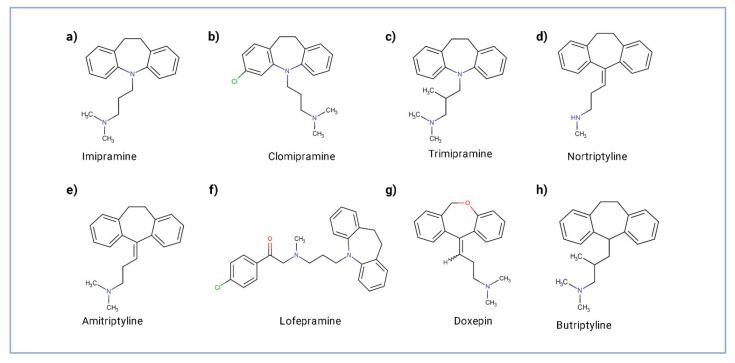
Structure of most common TCAs. (**a**–**c**) TCAs predominantly act by inhibiting the reuptake of serotonin; (**d**) nortriptyline predominantly by inhibiting reuptakes of norepinephrine; (**e**,**f**) balanced reuptake inhibitors of serotonin and norepinephrine; (**g**,**h**) balanced reuptake inhibitors of serotonin and norepinephrine with additional anti-H1 activity.

**Figure 2 cancers-14-03248-f002:**
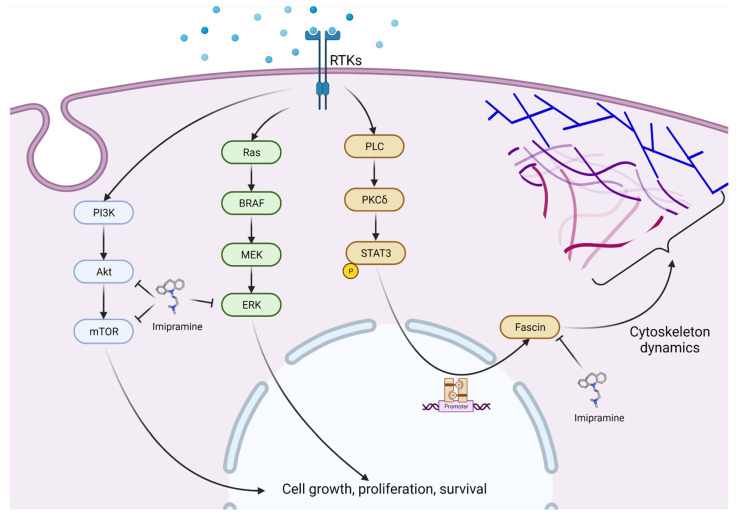
Relevant molecular pathways involved in the antitumoral mechanism of action of imipramine. Imipramine inhibits the phosphorylation of Akt (Ser473) and mTOR (Ser2481) in a time-dependent manner [57,80]. Imipramine is able to reduce the phosphorylation of ERK and P-65 NF-κB, leading to inactivation of ERK/NF-κB signaling transduction [55]. Imipramine can also inhibit the actin-binding protein fascin. This inhibitory effect is mediated by direct binding of imipramine to fascin [71]. Created with BioRender.com.

**Table 1 cancers-14-03248-t001:** Clinical trials with TCAs to treat neuropathic pain in oncology patients.

Clinical Trial ID Phase Status	Title	Conditions	Treatments	Primary Outcome Measures/Secondary Outcome Measures	Study Results/Publications
ISRCTN49116945 Completed	“A randomised, double-blind controlled trial of ketamine versus placebo in conjunction with best pain management in neuropathic pain in cancer patients”	Neuropathic pain related to cancer	Ketamine HCLD Placebo	SF-MPQ	Ketamine was equivalent to placebo for cancer-related neuropathic pain [48].
				VAS Patient distress EuroQol thermometer HADS Daily opioid requirement	
NCT00740571 Phase 3 Unknown	“Amitriptyline or Pregabalin to Treat Neuropathic Pain in Incurable Cancer”	Cancer Neuralgia	Amitriptyline Pregabalin	VAS score	No results available
				EQ-5D McGill pain questionnaire EORTC-C30 HADS	
NCT00471445 Phase 3 Completed	“Topical Amitriptyline and Ketamine Cream in Treating Peripheral Neuropathy Caused by Chemotherapy in Cancer Patients”	Neurotoxicity Pain Peripheral neuropathy Unspecified adult solid tumor	Ketamine/amitriptyline NP-H cream Placebo	Change in average daily peripheral neuropathy intensity score	The KA treatment showed no effect on 6-week CIPN scores [49].
NCT00516503 Completed	“Baclofen–Amitriptyline Hydrochloride–Ketamine (BAK) Gel in Treating Peripheral Neuropathy Caused by Chemotherapy in Patients with Cancer”	Lymphoid neoplasm Myeloid neoplasms Neurotoxicity Pain Unspecified adult solid tumor	Baclofen/amitriptyline/ketamine gel Placebo	EORTC–QLQ–CIPN20 to measure Total Sensory Neuropathy	Topical treatment with BAK–PLO appears to somewhat improve symptoms of CIPN. This topical gel was well tolerated, without evident systemic toxicity. Further research is needed with increased doses to better clarify the clinical role of this treatment in CIPN [50,51].
				EORTC–QLQ–CIPN20 to measure motor neuropathy and autonomic symptoms and functioning POMS BPI Peripheral Neuropathy Questionnaire Adverse event profile through clinical assessment by NCI CTCAE v3.0	
NCT00798083 Phase 3 Completed	“Neuropathic Pain Caused by Radiation Therapy”	Neuropathic pain secondary to radiation therapy	Topical amitriptyline 2%, ketamine 1%, and lidocaine 5% in PLO	UWNPS	No results available
				STAT	

BPI, Brief Pain Inventory; CIPN, Chemo-Induced Peripheral Neuropathy; EORTC-C30, European Organisation for Research and Treatment of Cancer; EORTC–QLQ–CIPN20 European Organization for Research and Treatment of Cancer–Quality of Life Questionnaire–Chemo-Induced Peripheral Neuropathy; EQ-5D, EuroQol 5 Dimension; EuroQol, European Quality of Life instrument; HADS, Hospital Anxiety and Depression Scale; HCLD, hydrochloride; KA, ketamine/amitriptyline; PLO, pluronic lecithin organogel; POMS, Profile of Mood States (McGill); NCI CTCAE v3.0, National Cancer Institute Common Terminology Criteria for Adverse Events; STAT, Skin Toxicity Assessment Tool; SF-MPQ, Short-Form McGill Pain Questionnaire; UWNPS, University of Washington Neuropathic Pain Scale; VAS, Visual Analogue Scale.

**Table 3 cancers-14-03248-t003:** Clinical trials evaluating antitumoral effects of TCAs.

Clinical Trial ID Phase Status	Title	Conditions	Treatments	Primary Outcome Measures/Secondary Outcome Measures	Study Results/Publications
NCT01719861 Phase 2 Terminated	“Phase 2a Desipramine in Small Cell Lung Cancer and Other High-Grade Neuroendocrine Tumors”	Small cell lung cancer Neuroendocrine tumors	Desipramine HCLD	ORR	No clinical or radiographic benefit was observed, so this trial was terminated [134].
				Desipramine maximum dose Median serum desipramine levels Median PFS Median OS	
NCT03122444 Early Phase 1 Recruiting	“Imipramine on ER + ve and Triple Negative Breast Cancer”	Breast cancer	Imipramine	Decrease in the proliferation rate of TNBC	No results available [111]
NCT04704453 Phase 2 Recruiting	“Study to Evaluate the Interest of Qutenza in Patients with Head and Neck Cancer in Remission and With Sequelae Neuropathic Pain”	Head and neck cancer	Capsaicin patch Amitriptyline	The rate of patients with a decrease in average pain	No results available
				NPSI questionnaire Adverse events evaluated by NCI-CTCAE V5 QLQ-C30.	
NCT02881125 Phase 1 Completed	“Paclitaxel and Nortriptyline Hydrochloride in Treating Patients with Relapsed Small Cell Carcinoma”	Small cell carcinoma	Nortriptyline HCLD Paclitaxel	Maximum tolerated dose.	No results available
				Response evaluation criteria in solid tumors v.1.1 OS PFS	
NCT04863950 Phase 2 Not yet recruiting	“Investigator-Initiated Study of Imipramine Hydrochloride and Lomustine in Recurrent Glioblastoma”	Glioblastoma	Lomustine Imipramine HCLD	PFS	No results available
EudraCT-2021-001328-17 Phase 2 Ongoing	“Histological and clinical effects of Imipramine in the treatment of patients with cancer over-expressing Fascin1”	Colorectal cancer and TNBC showing fascin overexpression.	Imipramine Placebo/neoadjuvance	Comparison of the histological traits of invasive tumor front of the surgical tumor resection specimen between the intervention group and the placebo group:(1) Fascin1 expression in tumor tissue. (2) Histological manifestations of the EMT.(3) Invasive histological manifestations. (4) Histological manifestation of the immune response. (5) EMT molecular manifestations.	No results available
				Monitoring of minimal residual disease, using circulating DNA	

EMT, epithelial–mesenchymal transition; HCLD, hydrochloride; NCI CTCAE, National Cancer Institute Common Terminology Criteria for Adverse Events; NPSI, Neuropathic Pain Symptom Inventory; ORR, overall response rate; OS, overall survival; PFS, progression-free survival; QLQ, Cancer Quality of Life Questionnaire; TNBC, triple-negative breast cancer.

## Data Availability

No new data were created or analyzed in this study. Data sharing is not applicable to this article.

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
