# Peer review of "Antitumoral Effects of Tricyclic Antidepressants: Beyond Neuropathic Pain Treatment"

_cancers, 2022, doi:10.3390/cancers14133248_

Round 1

Reviewer 1 Report

In the present review the authors gave a broad report about antitumoral effect of trycyclic antidepressants. Although they made a systemic review, some parts of the manuscript should be changed to be acceptable for publication.

The major limitation of the review is part 3 ”In vitro and pre-clinical evidence of antitumoral properties of tricyclic antidepressants” that is little bit confusing and in some part hard to follow. It’s not clear which examples are listed in Table 2 and which are described in the main text. For instance, examples about hepatocellular carcinoma, osteosarcoma and multiple myeloma are described only in the text, while most of examples about lung carcinoma, prostate carcinoma and breast carcinoma are given only in Table 2. In my opinion all examples the authors referred to should be clearly described in the text emphasizing molecular mechanism of drug in specific cancer type, its in vivo studies and then studies with cancer patients.

In the paragraph 137 the sentence “found a statistically significant 45% reduction in the risk of colorectal cancer for regular use of SSRIs” is the same sentence that Coogan et al., 2009 used in Discussion of their manuscript. Although the authors cited Coogan et al., 2009, they should avoid using the same sentences from other manuscripts to evade plagiarism.

The authors should use only one synonym throughout the text, noradrenaline or norepinephrine, not both.

Author Response

We want to thank the reviewer for providing comments that has improved the manuscript´s quality. We proceeded to answer all points raised:

Reviewer (R). The major limitation of the review is part 3 ”In vitro and pre-clinical evidence of antitumoral properties of tricyclic antidepressants” that is little bit confusing and in some part hard to follow. It’s not clear which examples are listed in Table 2 and which are described in the main text. For instance, examples about hepatocellular carcinoma, osteosarcoma and multiple myeloma are described only in the text, while most of examples about lung carcinoma, prostate carcinoma and breast carcinoma are given only in Table 2. In my opinion all examples the authors referred to should be clearly described in the text emphasizing molecular mechanism of drug in specific cancer type, its in vivo studies and then studies with cancer patients.

Author´s response (AR): The reviewer´s comment is pertinent and both table and text have been modified in order to present the information in a clearer and coordinated way. Thus, cancer types cited in the text have been included in the table. Drugs´ mechanisms have now been emphasized and described when known and the text order has now been established following the reviewer´s suggestion.     

R: In the paragraph 137 the sentence “found a statistically significant 45% reduction in the risk of colorectal cancer for regular use of SSRIs” is the same sentence that Coogan et al., 2009 used in Discussion of their manuscript. Although the authors cited Coogan et al., 2009, they should avoid using the same sentences from other manuscripts to evade plagiarism.

AR: The text has been modified accordingly

R: The authors should use only one synonym throughout the text, noradrenaline or norepinephrine, not both.

AR: The term "norepinephrine" has been chosen and so it has now been used throughout the whole text.  

Reviewer 2 Report

The manuscript is a well-written review of the potential of tricyclic antidepressants (TCAs) as anticancer drugs. In their manuscript, the authors nicely described and updated in vitro and in vivo results showing TCAs' antiproliferative and pro-death activity in several tumor models.

Strengths: this review addresses an interesting and relevant topic in cancer, such as the repurposing of already used and safe drugs for antitumor treatments. The manuscript is well structured, and the review process appears to be performed with rigor.

Limitations: mechanisms by which these drugs induce tumor cell death are unclear. Moreover, in vitro results are not supported by the results of clinical trials.

Minor comments

lines 198-203:

In section 2, the authors describe the role of TCAs in cancer-related neuropathic pain; for this reason, this sentence may fit better in the context of section 3.

Graphical Abstract:

The arrows indicating in vitro evidence, in vivo evidence, and clinical trials should be moved to the right (near antitumoral activity).

 Figure 2:

The authors should describe, whether it is known,  the mechanisms by which imipramine blocks AKT, mTOR, and Fascin.

Lines 307-314:

the sentence is unclear; rephrase.

Author Response

We want to thank the reviewer for its critical revision of the manuscript. We have given point-by-point response to his/her comments with the aim of improving the manuscript´s quality. 

Reviewer (R): The manuscript is a well-written review of the potential of tricyclic antidepressants (TCAs) as anticancer drugs. In their manuscript, the authors nicely described and updated in vitro and in vivo results showing TCAs' antiproliferative and pro-death activity in several tumor models.

Strengths: this review addresses an interesting and relevant topic in cancer, such as the repurposing of already used and safe drugs for antitumor treatments. The manuscript is well structured, and the review process appears to be performed with rigor.

Author´s response (AR): We thnk the reviewer for these positive feedback.

R: Limitations: mechanisms by which these drugs induce tumor cell death are unclear. Moreover, in vitro results are not supported by the results of clinical trials.

AR: The comment is pertinent. Drugs´ mechanisms, when known, have now been added in the text. In order to reflect the limitation that in vitro results still, not strongly support the results of the clinical trial, the following paragraph has been added;   

  1. Limitations of the study

The mechanisms by which TCAs induce tumor cell death seems to be heterogeneous and dependent on tumor type and even subtype. Given this fact and that of the scarse number of clinical trials evaluating the antitumoral effects of TCAs, it is not surprising that the promising results obtained in in vitro and  in vivo preclinical  studies are not supported by the clinical results. In fact, one phase 2a clinical trial using desipramine in small cell lung cancer and other high-grade neuroendocrine tumors were terminated after finding no clinical or radiographic benefit. Despite this fact, new clinical trials are being initiated in different cancer types evaluating the effect of various TCAs. More basic and translational research will be needed to gain a more detailed understanding of the mechanisms of action and to validate molecular targets of TCAs. As this knowledge continues growing, the selection of patients according to specfic histological or molecular features seems crucial for finding the benefit of TCAs as antitumoral agents.

R: Minor comments

lines 198-203:

In section 2, the authors describe the role of TCAs in cancer-related neuropathic pain; for this reason, this sentence may fit better in the context of section 3.

Graphical Abstract:

The arrows indicating in vitro evidence, in vivo evidence, and clinical trials should be moved to the right (near antitumoral activity).

Figure 2:

The authors should describe, whether it is known,  the mechanisms by which imipramine blocks AKT, mTOR, and Fascin.

Lines 307-314:

the sentence is unclear; rephrase.

AR: All these four minor points have been addressed and corrected accordingly.

Round 2

Reviewer 1 Report

The authors accepted all my suggestions and I recommend this manuscript for publication in Cancers.